# Novel Approaches to Possible Targeted Therapies and Prophylaxis of Uterine Fibroids

**DOI:** 10.3390/diseases11040156

**Published:** 2023-11-01

**Authors:** Maria V. Kuznetsova, Narine M. Tonoyan, Elena V. Trubnikova, Dmitry V. Zelensky, Ksenia A. Svirepova, Leila V. Adamyan, Dmitry Y. Trofimov, Gennady T. Sukhikh

**Affiliations:** 1Kulakov National Medical Research Center of Obstetrics, Gynecology and Perinatology, 117997 Moscow, Russia; mkarja@mail.ru (M.V.K.); tonnar.13@bk.ru (N.M.T.); k_svirepova@oparina4.ru (K.A.S.); l_adamyan@oparina4.ru (L.V.A.); d.trofimov@dna-technology.ru (D.Y.T.); gtsukhikh@mail.ru (G.T.S.); 2Genetics Research Laboratory, Kursk State University, 305000 Kursk, Russia; 3Valuiky Central District Hospital, 309996 Valuiky, Russia; dmitriizelenskii@mail.ru

**Keywords:** uterine leiomyoma, fibroid, pregnancy, therapy, targets, inhibitors, vaccines, humanized antibodies, *MED12*, *HMGA2*, fumarate hydratase, *IL17B*, *MMP11*, *MMP16*

## Abstract

Uterine leiomyomas are the most common benign tumors in women of childbearing age. They may lead to problems of conception or complications during the gestational period. The methods of treatment include surgical (myomectomy and hysterectomy, embolization of arteries) and therapeutic treatment (ulipristal acetate, leuprolide acetate, cetrorelix, goserelin, mifepristone). Both approaches are efficient but incompatible with pregnancy planning. Therefore, there is a call for medical practice to develop therapeutical means of preventing leiomyoma onset in patients planning on becoming pregnant. Based on the analysis of GWAS data on the search for mononucleotide polymorphisms associated with the risk of leiomyoma, in meta-transcriptomic and meta-methylomic studies, target proteins have been proposed. Prospective therapeutic treatments of leiomyoma may be based on chemical compounds, humanized recombinant antibodies, vaccines based on markers of the uterine leiomyoma cells that are absent in the adult organism, or DNA and RNA preparations. Three different nosological forms of the disease associated with driver mutations in the *MED12, HMGA2*, and *FH* genes should be considered when developing or prescribing drugs. For example, synthetic inhibitors and vaccines based on matrix metalloproteinases *MMP11* and *MMP16* are expected to be effective only for the prevention of the occurrence of *MED12*-dependent nodules.

## 1. Introduction

Uterine fibroids are most common in women aged 35–50 years (frequency of 30–35%), but there is a tendency for the first onset of the disease at an earlier age—25 years and even younger [1]. In the structure of gynecological morbidity, uterine fibroids occupy second place in the frequency of occurrence, second only to inflammatory diseases of the female reproductive system, and remain the most common benign tumors in gynecology [2].

According to data from various sources, among women of reproductive age, the incidence of the disease varies from 30 to 50%. The average age of detection of uterine fibroids is 32.8 ± 0.5 years, and indication for surgical treatment occurs after 4.4 ± 0.3 years [3]. In 20–30% of women, fibroids are diagnosed at their reproductive age, and in 30–40% of women over 40 years old. The true frequency of the disease cannot be determined unambiguously due to the fact that >30% of patients have uterine fibroids without clinical manifestations. Leiomyoma is diagnosed in adolescents with uterine bleeding in 5–7%, in 4%—in the age group from 20 to 30 years, in 11–18%—in 30–40 years, in 33–40%—in 40–60 years. In recent years, there has been a tendency toward a decreasing average age when the disease is detected for the first time. There is a category of women aged 20–25 years suffering from this pathology, which almost did not exist formerly [4]. This phenomenon is associated with a decrease in the average age of menarche achievement. There are no cases of fibroids in girls before puberty. There has been a decrease in cases of the disease in the menopausal period, which may be due to a gradual increase in the average age of menopause, which has increased from 40 to 50–52 years over the past 200 years [3]. It is also known that the symptomatic leiomyoma requiring treatment is more often manifested in perimenopause, while after menopause, its frequency decreases sharply [5]. Women who have given birth to five children are four times less at risk of the disease compared to those who have not given birth [6]. In primiparous women over 30 years of age, fibroids occur in 15–17% [3].

Among women diagnosed with uterine leiomyoma, 15–30% have symptoms of varying severity, including pain syndrome, infertility, dysfunction of adjacent organs, abnormal uterine bleeding, anemia, and a number of other severe complications [7]. Despite their benign nature, leiomyomas are able to metastasize, penetrating into various tissues of the body outside the uterus, in particular, into the lungs [8].

In gynecological hospitals in Russia, up to 50–70% of operations are performed specifically for symptomatic uterine leiomyoma. In addition, 40 to 60% of hysterectomies are performed specifically for this disease. In European countries, more than 300,000 surgical interventions related to uterine fibroids are performed annually [9], and in the USA—approximately 200,000 myomectomies and 30,000 hysterectomies [10]. In the USA, approximately 600,000 hysterectomies are performed every year, 200,000 of which are due to fibroids, beyond 30,000 myomectomies [11]. In the European Union, the yearly number of hysterectomies reaches more than 300,000, and in China—1 million [12]. In Canada, every fourth woman above 45 has been subjected to a hysterectomy, in the UK—every fifth, in the USA—every third (>80% of women are under 49 and 50% of them are under 40) [13]. In [14], there is an estimate that uterine fibroids affect up to 77% of women during menopause, and the annual health care costs for combating this disease in the United States amount to USD 34 billion.

Fibroids are capable of rapid growth during pregnancy, which can lead to violations of the normal growth of the placenta and the development of defects in the fetus. Often, such problems might lead to the performance of urgent myomectomies in the early stages of pregnancy or cesarean section before the delivery date. Fibroids are able to grow rapidly in size during pregnancy, and in the case of subserous localization, tumors form adhesions with internal organs, which cause serious difficulties for uterine contractions during and after childbirth. In addition, there are cases of postpartum bleeding due to uterine atony [15].

The development of postoperative relapses is a separate problem in the treatment of the disease, since the recurrence rate after myomectomy reaches 90%, while in almost every fourth case there is a need for repeated treatment until a new surgical intervention. The probability of recurrence depends on how thoroughly all the existing fibroids were removed; however, in the presence of multiple tumors, the risk of recurrence is higher, since often small fibroids are not visualized and, as a result, are not removed, continuing to grow in the postoperative period. According to the review by [16], the risk of repeated surgery for multiple uterine fibroids is 26%. With a single fibroid, recurrence of the condition occurs in 27%, and the risk of repeated surgery due to recurrence is 11% [17].

Hysterectomy is a radical way to prevent the recurrence of leiomyoma; however, it is not acceptable for women planning pregnancy. Moreover, a significant problem arising as a result of a myomectomy is the traumatization of the uterus, the occurrence of scars that lead to the disruption of its normal functioning during pregnancy, and the risk of ruptures during stretching and contraction before and during the course of childbirth. In addition, the growth rate of fibroids accelerates during pregnancy, which is usually explained by a high level of blood progesterone [18]. Thus, the most significant medical and social problem of fibroid therapy is the development of treatment methods that prevent the recurrence of the disease in patients planning pregnancy or in the process of gestation. The purpose of this review is to systematize existing methods for preventing the recurrence of fibroids or slowing the growth of existing fibroids in patients planning childbirth or having a pregnancy.

## 2. Review Structure and Design

We suggest specifying the following principal directions of uterine leiomyoma etiology studies and the development of approaches to the treatment of this condition: (1) *HMGA2* gene overexpression as a driver mutation in UL; (2) the driver mutations in the *MED12* gene in UL; (3) the driver null-mutations in the *FH* gene in UL; (4) using the agonists and antagonists of steroid hormones as UL growth suppressors; (5) PI3K/Akt/mTOR signaling pathway in UL and its inhibitors as UL growth suppressors. The first three directions are important since ULs with different driver mutations differ substantially in the expression profile, and consequently, in the molecular targets of the candidate therapeutics. Direction 5 allows for analyzing the efficacy and side effects of commercially available and prospective medicines used for UL treatment. Directions 5 and particularly 6 allow for identifying novel molecular targets for UL drug and vaccine design that are prerequisites for fibroid formation and growth but absent in the normal myometrium. The key publications of each direction are summarized in Table 1.

Using the cited publications, the following sections are formed in this review: Section 3: Mechanisms and risk factors for the development of recurrent uterine leiomyoma; Section 4: Existing methods of medicinal treatment of the recurrent uterine leiomyomata; Section 5: Prediction of genetic risk and identification of potential therapeutic targets in leiomyoma cells using metagenomic, metatranscriptomic, and metamethylomic methods of analysis of complete genomes; Section 6: Approaches to development of new candidate therapeutics for the prevention and treatment of relapses of leiomyoma, including patients preparing for pregnancy or in the process of gestation; and Section 7: Conclusions.

## 3. Mechanisms and Risk Factors for the Development of Recurrent Uterine Leiomyoma

Currently, the molecular mechanism of the occurrence of the fibroids remains unclear [57]. It is assumed that the initiator of the transformation of myometrial cells or their stem progenitors into fibroids cells is local hypoxia and the effect of hormones, the concentration of which varies significantly depending on the phase of the estrous cycle [58]. There is no doubt about the dependence of the occurrence and growth of fibroids on sex steroids—estrogens, progesterone, and pituitary hormones—gonadotropin, follicle-stimulating, luteinizing and anti-Mullerian hormones, prolactin. An essential role is played by the individual genotype of the patient (heredity), ethnic origin, and a high body mass index.

In 2017, the concept of a driver mutation leading to the appearance of a fibroid was established in the literature [30]. It is noteworthy that each fibroid is monoclonal, originating from a single progenitor cell [59]. This leads to the fact that the genotype of the so-called “driver mutation” in different fibroids of one patient with multiple fibroids usually does not match. The most common driver mutations are somatic missense mutation in exon 2 of the *MED12* gene (codons 130 and 131 and in frame deletions) encoding the regulatory subunit of RNA polymerase 2, and rearrangements leading to overexpression of the *HMGA2* protein, a non-histone chromatin-binding protein involved in the regulation of the chromatic condensation and regulation of gene expression. These driver mutations account for more than 95% of cases of leiomyoma. In addition, biallelic inactivation of the fumarate hydratase gene (Krebs cycle enzyme) and rare chromosomal rearrangements, in particular, the loss of the Xq22.3 region, leading to the loss of COL4A5 - COL4A6 collagen genes, can act as a driver mutation [60].

Conflicting opinions have been expressed in the literature regarding the possibility of two driver mutations co-existing in one cell. Most authors suggest that two mutations in one cell are incompatible. In works [57,61,62,63], it was shown by methods of genome-wide metatranscriptomic and metamethylomic analysis that the global profiles of gene expression and methylation in the case of *MED12*-dependent and *HMGA2*-dependent fibroids differ significantly. A feature of *MED12*-dependent fibroids is a high proportion of fibroblasts (which do not have a driver mutation and correspond to the normal genotype of the patient), a high content of extracellular matrix (ECM), and a low degree of vascularization. The fibroids of this type have a relatively small size but show a tendency of multiplicity. Due to the high content of fibroblasts, the growth of *MED12*-dependent fibroids is stimulated not only by progesterone, but also by estrogens, which is not typical of other fibroid types.

*HMGA2*-dependent fibroids do not contain a clearly determined *sensu stricto* driver mutation, but their common and fundamental feature is the overexpression of the *HMGA2* gene, and sometimes also of the HMGA1 and PLAG1 genes, which determine the overall expression profile and metabolism of the cell. The most common cause of *HMGA2* overexpression may be the translocation of the 12q14.3 region containing the *HMGA2* gene to chromosome 14 or to other parts of the genome, where the gene falls under the control of promoters of other genes (most often RAD51L1), or its 3′-terminal region is deleted, contributing to the destabilization of the transcript. The second common cause of *HMGA2* overexpression is deletion of the long arm of chromosome 7 (q21.2q31.2), which contains the regulatory RNA genes miR-21, miR-23b, miR-29b, and miR-197 [64] and the CUX1 homeobox gene [27]. These micro-RNAs, primarily miR-29b, are repressors of *HMGA2* expression, so their loss leads to overexpression of the regulated gene. More than half of the fibroids with *HMGA2* overexpression do not have detected chromosomal rearrangements, and *HMGA2* overexpression is usually suggested to be achieved there by changing the methylation status [65]. The presence of several independent mechanisms of *HMGA2* overexpression significantly complicates the identification of such fibroids at the level of laboratory diagnostics; the most reliable way to solve this problem is to detect the *HMGA2* protein using immunohistochemical staining or its transcript with RT-qPCR. The accuracy of diagnosis is facilitated by the absence or extremely low level of *HMGA2* expression in normal myometrium, although there are reports of the possibility of moderate *HMGA2* expression in *MED12*-dependent fibroids.

Biallelic inactivation of the fumarate hydratase gene is characteristic mainly of a rare type of leiomyoma with bizarre nuclei [66]. According to [62], cells of this type have a special type of metabolism: the overexpression of genes of the pentose phosphate pathway of sugar oxidation and glucose-6-phosphate dehydrogenase, suppression of the expression of Krebs cycle enzymes and pyruvate dehydrogenase, and the hyperproduction of enzymes responsible for the synthesis of reduced glutathione.

Other driver mutations occurring in the fibroids are so rare that their systematic study on the basis of statistically reliable data is practically difficult. Information about these mutations is present in the works by [60,62]. It was suggested that the driver of deletions in the COL4A5 and COL4A6 genes encoding basal membrane collagens is the IRS4 gene, the product of which is an intracellular messenger phosphorylated by the insulin receptor or IGF1 after binding to them by an extracellular ligand [65]. In this regard, one should note that the IRS4 gene is located in the Xq22.3 chromosome region nearby to the COL4A5 and COL4A6 genes. Holzmann et al., 2014 [28] reported that leiomyoma foci without somatic mutations in the *MED12* gene nor overproducing *HMGA2* are prone to so-called chromotripsis (multiple duplications and deletions of small segments grouped mainly into five chromosomal regions 2p14 - 2pter, 2q33.1 - 2q37.3, 5q31.3 - 5qter, 11q14.1 - 11qter, and 18p11.21 - 18q2.3). Due to the small size of the rearranged fragments of genomic DNA, such rearrangements are difficult to detect by methods of classical cytogenetics.

## 4. Existing Methods of Medicinal Treatment for Recurrent Uterine Leiomyomata

Historically, the first methods of therapy for leiomyomas were the use of hormone antagonists or antitumor cytostatics acting at the level of the PI3K (phosphatidylinosylkinase 3)/Akt/mTOR signaling pathway (target of the antibiotic rapamycin inhibiting autophagy) [55]. This is due to the fact that the hormonal dependence of leiomyoma has long been known, and antitumor cytostatics are actively used in practice to suppress the growth of malignant neoplasms. The desire to use the PI3K/Akt/mTOR pathway as a therapeutic target is due to the fact that its shutdown does not cause immediate cell death, but leads to the so-called phenomenon of oncogene-induced senescence (OIS), when cells under the action of proteins p16, p19, p53, and p21 gradually lose their division potential and weaken the antioxidant protection system, which weakens the tumor aggressiveness. The induction of this pathway is the normal mechanism of excessive myometrium and degeneration after the childbirth, and its launching in leiomyoma cells theoretically can be highly specific and cause a relatively low level of side toxicity.

A consistent increasing production of pACT and GF2BP2 was observed in the cells of the fibroids with an overexpression of *HMGA2* [54]. The specific suppression of GF2BP2 expression by RNA interference causes a decrease in the level of pACT.

The suppression of pACT expression with miR-182 micro-RNA or with a chemical inhibitor MK-2206 was reported to lead to an increase in the production of proteins p16 (Ink4a), p53, and p21, which ultimately causes the phenomenon of premature senescence in leiomyoma cells [51]. However, [26] reported that basic expression levels of p16 (Ink4a) and p19 (Arf) in leiomyoma cells was many times higher compared to the myometrium, but this fact does not lead to signs of oncogen-induced senescence of the fibroids.

Compound T-5224, an inhibitor of transcription factor AP1 (activating protein 1) involved in the activation of ECM synthesis and the formation of fibrosis foci, was described by [52]. Pilgrim et al., 2020 [56] reported that after the stimulation of a leiomyoma cell culture with TGFß3 for 24 h, the content of fibronectin increased by 2.16 ± 0.14 times, and with versican (specific hyaline cartilage hyalurone-binding glycoprotein), by 4.71 ± 0.15 times. The contents of collagen 1A after 6 h increased by 1.32 ± 0.01 times compared to the baseline level, and after 24 h—by 6.49 ± 0.02 times. At 4 h after treatment with an AP1 inhibitor (SR11302), a 0.59 ± 0.03-fold decrease in the content of collagen 1A was observed, and after 6 h—a 0.42 ± 0.05-fold decrease. The content of versican under the action of SR11302 decreased by 0.84± 0.04 times 6 h after treatment. The inhibitor significantly reduced the fibronectin content after 8 h of treatment (by 0.68 ± 0.05-fold).

The use of gonadotropin-releasing hormone (GnRH) agonist trials (leuprolide acetate) and GnRH antagonists (cetrorelix acetate) as a candidate therapy for leiomyomas is described by [42]. It was reported that 3D cultures of leiomyoma cells exposed to estrogen E2 for 24 h showed increased expression of collagen 1, fibronectin, and versican, which persisted for 72 h. Progesterone treatment increased the level of collagen 1 within 24 h after exposure. The simultaneous application of estrogen and progesterone caused a significant increase in all ECM proteins. When treating the cultures of leuprolide acetate and cetrorelix acetate for 24 h, a significant decrease in the production of ECM proteins was observed. Both compounds reduced the production of ECM proteins both in the absence and presence of one or both sex steroids. There are some other communications about GnRH agonist and antagonist trials for uterine fibroid therapy, e.g., linzagolix (OBE 2109, KLH 2109) [45], relugolix as a separate substance or in combination with estradiol and norethindrone acetate (NETA) [47], elagolix [49], and ganirelix [48].

The use of selective progesterone receptor modulators (ulipristal acetate and asoprisnil), antiprogestin (mifepristone—RU486) is described in [43]. The treatment of immortalized two-dimensional (2D) and three-dimensional (3D) human leiomyoma and myometrial cells with progesterone agonist progestin stimulated the production of COL1A1, fibronectin, versican, and dermatopontin. Treatment with mifepristone, an approved agent for the treatment of progesterone-dependent types of breast cancer, suppressed the synthesis of ECM components, especially versican. The combined treatment of cultures with an agonist and an antagonist of progesterone caused suppression of the synthesis of ECM components. In the countries of the European Union and Canada, ulipristal acetate (SPRM) is allowed for medical use as a contraceptive and in the treatment of uterine fibroids and endometriosis in doses from 5 to 10 mg. Asoprisnil (J867), a compound of the same group as ulipristal acetate, is also a selective modulator of progesterone receptors (SPRM), exhibiting the properties of both an agonist and an antagonist, depending on the type of a target tissue.

The effects of goserelin, an agonist of luteinizing hormone-releasing hormone, on serum leptin levels and body composition in women with uterine leiomyoma were reported by [39]. Fifteen women with a normal course of the sexual cycle participated in the trials. The serum concentrations of leptin, insulin, testosterone, progesterone, and estradiol were measured in all subjects, as well as their body mass index and waist–hip ratio before and after 4, 8, and 12 weeks of treatment with goserelin, which was given at a dose of 3.6 mg once every 4 weeks. Body fat mass and muscle mass were measured by two-energy radiographic densitometry at the beginning and after 12 weeks of therapy. The treatment led to a significant regression of fibroids. Body weight, fat, and muscle mass did not change. During the treatment, there were no changes in the level of leptin in the blood plasma. The level of estradiol in the blood plasma decreased below the level typical for postmenopause. Progesterone in the plasma decreased significantly, and testosterone tended to decrease throughout the study.

The results of clinical trials for the treatment of uterine leiomyoma with *Clostridium histolyticum* collagenase (CCH), which selectively hydrolyzes collagen types I and III and changes the stiffness of the ECM, in combination with verteporfin, an inhibitor of the YAP factor (Hippo/YAP pathway) and the antifibrosis drug nintedanib were reported in [44]. The introduction of CCH in doses of 0.1–0.2 mg/cm^3^ into fibroids after 60 days resulted in a 46% decrease in stiffness compared to the control. The level of PCNA, a marker of cell proliferation of nuclear antigen of proliferative cells, was reduced 60 days after the injection of high doses of CCH. The contents of key intracellular growth signaling factors Hippo and the phosphorylated form of YAP (p-YAP), as a result of the use of CCH, were increased, which contributed to maintaining a high rate of fibroid growth. The inhibition of YAP by verteporfin reduced cell proliferation, gene expression, and the proteins of key factors contributing to fibrosis and mechanotransduction in fibrous cells. The antifibrotic drug nintedanib additionally reduced the activity of YAP and showed an antifibrotic effect.

Trials of the plant-derived polyphenolic compound resveratrol for the treatment of leiomyoma are described in [67]. The authors reported a clear inhibitory effect of resveratrol on the proliferation of primary cultures of human uterus leiomyoma cells. Resveratrol stopped cell proliferation via avß3 integrin signaling since expression of this protein was suppressed by the drug. At the same time, resveratrol inhibited the constitutive phosphorylation of AKT in fibrous cells. Resveratrol treatment induced the expression of pro-apoptotic genes: cyclooxygenase (COX)-2, p21 and CDKN2. On the contrary, the expression of proliferative (anti-apoptotic) genes was either suppressed (BCL2), or unchanged (cyclin D1 and PCNA). There was a decrease in the production of insulin-like growth factor receptor (IGF-1R). Resveratrol treatment suppressed IGF1-induced proliferation of the uterine leiomyoma cells. The authors suggested that the arrest of leiomyoma cell growth by resveratrol occurs as a result of the cross-interaction between integrin avß3 and IGF-1R. This report is promising, but it should be borne in mind that, currently, resveratrol is not available for pharmaceutical use since its chemical synthesis is not established, and its content in plant raw materials is not sufficient for practical use. Moreover, clinical trials of this drug have not yet begun, which does not allow for expecting its rapid introduction into practical use.

A method of therapy for leiomyomata with a driver mutation in the fumarate hydratase gene was proposed in [37]. They claimed that fumarate and succinate inhibit the DNA repair mechanism that involves the homologous recombination mechanism and is necessary to eliminate double-stranded breaks in chromatin. This circumstance makes tumor cells, unlike normal ones, vulnerable to olaparib and niraparib, which are synthetic inhibitors of poly-ADP-ribose polymerase (PARP1, also known as NAD+-ADP-ribosyltransferase 1 and PARP2) that bind at the NAD+-binding site of the target enzyme. PARP inhibitors are approved for pharmaceutical use and should be suggested as available potential medicines for atypical variants of leiomyoma containing a mutation of biallelic inactivation of the fumarate hydratase gene.

## 5. Prediction of Genetic Risk and Identification of Potential Therapeutic Targets in Leiomyoma Cells Using Metagenomic, Metatranscriptomic, and Metamethylomic Methods of Analysis of Complete Genomes

To date, a number of works have been carried out worldwide that are aimed at identifying factors of hereditary predisposition to the occurrence of leiomyomas [14,68,69,70,71,72,73]. The main directions of the studies are as follows: (1) genome-wide associated studies pursuing mapping polymorphisms associated with predisposition to UL onset; (2) genome-wide transcriptome and DNA-methylome studies of fibroids, pursuing the discovery of the gene transcription alterations leading to the transformation of the normal myometrium to the fibroid tissue; (3) candidate medicines for UL treatment based on small molecules, vaccines, and RNA are summarized in Table 2. As a result, more than 100 single nucleotide polymorphisms (SNP) were identified, the genotype of which affects the risk of developing leiomyoma. However, the practical use of the results of these works is not organized so far. Moreover, although many of these studies used highly representative groups of several tens of thousands of patients and up to 523,000 control patients, none of the authors used data on the presence of a burdened family history of patients. Moreover, the tumor samples themselves were not classified depending on the type of driver mutation. Meanwhile, it can be assumed that the hereditary factors of predisposition to the occurrence of somatic mutations in the *MED12* gene, HGMA2 overexpression, and biallelic inactivation of the fumarate hydratase gene are unlikely to completely coincide. Therefore, now there is no possibility of predicting the risk of each type of leiomyoma based on the analysis of the patient’s genotype.

There are also a number of papers in the literature focused on the molecular profiling of leiomyoma at the level of transcriptome and DNA-methylome analysis [57,61,62,63]. Authors [57,62] classified the samples used in their works in accordance with the type of driver mutation. The data obtained allow us to draw a number of conclusions regarding the features of gene expression regulation that are important for the development of new leiomyoma therapies.

The work by [62] confirms the previously proposed hypothesis about a significant change in the expression of genes involved in the Wnt/β-catenin signal pathway: in all types of leiomyoma cells, these genes are suppressed. This is caused by an increase in the expression level of antagonists of Wnt—WIF1 and SFRP1, as previously described by [50]. Significantly, this change concerns both types of leiomyomas: those with somatic mutations in *MED12* and those with an overexpression of *HMGA2*. This conclusion is fully supported by the work of [57]. Therefore, secretory proteins Wif1 and Sfrp1 can be considered promising targets for removal using therapeutic antibodies that are universal for all leiomyoma types. The pituitary hormone prolactin is another promising target for removal from the organism in order to prevent relapses of uterine leiomyoma. According to [62], prolactin levels are steadily increased in all leiomyoma types, regardless of the driver mutation. The presence of prolactin in the mother’s body is critically important only during lactation, and therefore, the use of prolactin inhibitors, including those based on recombinant humanized antibodies at the stage of pregnancy has the prospect of achieving a good effect in terms of preventing the growth of fibroids without causing damaging side effects to the mother’s body and fetus. Maekawa et al., 2022 [57] also found that the demethylation of genes was associated with nucleosome assembly and telomerase activity (HIST1H4J, HIST1H4K, HIST1H4F) in all types of the uterine leiomyomata. In contrast, genes of inflammatory response (CCL2, AOX1, ACKR1), apoptosis (ANXA1, CITED2), and metabolism associated with reactive oxygen species undergo hypermethylation in all leiomyoma types. This information is important from the point of view of understanding the mechanisms of pathogenesis; however, it does not provide the key to choosing a target for therapy since most of these genes are active in normal tissues and cannot be switched off without severe damage to the organism. In a separate direct experiment using RT-qPCR, [57] demonstrated that 80–90% of fibroids of all types showed overexpression of the SATB2 and NRG1 genes compared to the myometrium, and the excess level varied from 1.5 to 20–30 times. Cell lines overexpressing SATB2 have a morphologically unusual cell type: they lose the fusiform shape peculiar for smooth muscle tumors and become similar to fibroblasts with elongated pseudopods. This suggests that SATB2 and NRG2 play an important role in the initiation of leiomyoma. We believe that these data allow us to consider the proteins Satb2 and Nrg1 as promising targets for the creation of therapies for all leiomyoma types.

More than 40 genes are listed in [57,61,62], the products of which are represented in fibroids of the *MED12*-dependent and *HMGA2*-dependent types at the level, significantly exceeding those in the normal myometrium. They should be considered promising targets for therapy. However, for the experimental verification of the effectiveness of candidate therapeutic agents, it is advisable to narrow down the fairly wide range of these targets, giving preference to two categories of proteins:Secreted proteins that do not remain on the surface of the cells producing them;Proteins present exclusively in leiomyoma cells and absent in any cells of normal tissues.

Such an approach to the selection of targets is dictated by the need for minimizing the side toxicity of candidate drugs, which can be achieved only if normal organs and tissues remain unaffected. In the first case (the secreted molecules), the proteins themselves and not the producer cells are targeted, and the therapy leads only to temporarily reducing their concentration in the extracellular space, while the producer cells remain alive. Therefore, requirements for the absence of the targets in normal cells may be not absolute. As for proteins present strictly in leiomyoma cells, they are found mainly among embryonic proteins that are absent in adult cells, but they often act as cancer markers. After sorting the potential targets according to the proposed principle, the following list of genes (proteins) can be presented:3.From the work by [61]: *PCP4*—Purkinje cell protein 4, expressed in embryogenesis; CHRDL2—growth factor, a chordin-like antagonist of BMP; RPE65—retinol transporter from the retinal epithelium; *MMP11*—matrix metalloproteinase 11 (or stromelysin 3), expressed in embryogenesis, causes metastasis of breast tumors; *MFAP2*—microfibrillar-associated protein 2, affects the motility of cancer cells in gastric cancer, regulates the expression of a5ß1 integrin via the ERK1/2 pathway.4.From the work by [62]:For *HMGA2*-dependent fibroid types: *HMGA2*, *GRPR*—gastrin precursor; *PLAG1*—proto-oncogene, the main participant of *HMGA2*-dependent signaling; *PAPPA2*—pappalysin, a protease that destroys IGFBP5, a marker of osteoblasts; *MB21D2*—nucleotidyltransferase, an anti-virus protection gene induced by interferon;For *MED12*-dependent fibroid types: *ADAM12*—membrane-bound metalloproteinase responsible for shedding—cleavage of extracellular receptor domains and proteoglycans; *MMP11*—matrix metalloproteinase 11 (or stromelysin 3), expressed in embryogenesis, causes metastasis of breast tumors; *MMP16*—matrix metalloproteinase 16, has a transmembrane domain, expressed in embryogenesis; *RAD51B*—protein of the DNA reparation system, similar to RecA, with ATPase activity, not expressed in normal tissues; *PCP4* is a protein of 4 Purkinje cells, expressed in embryogenesis; *RUNDC1* is a DNA-binding protein with a RUN-type domain at the C-terminus; *THSD4* (Adamtsl6-β) matrix metalloproteinase with a thrombospondin domain associated with the cell surface; participates in the formation of ECM microfibrils, ensuring elasticity and release of deposited growth factors. Adamtsl6-β expression increases during the formation of the periodontal ligament, which consists of a fibrillin-1 microfibril.

An Interesting finding by [61] is the detection of an increased level of *IL17B* mRNA in leiomyoma. *IL17B* is a cytokine derived from T-lymphocyte and involved in the initiation and maintenance of inflammation, fibrosis, and keratinization. An advantage of *IL17B* as a target for leiomyoma therapy is the availability of recombinant humanized antibodies approved for the treatment of psoriasis and psoriatic arthritis [81]. These drugs demonstrate moderate side effects, which facilitate their clinical trials.

The protein retinol transporter from the retinal epithelium (*RPE65*), which was identified in the same work as a marker of differential expression in leiomyoma, may also be considered a target available for therapy with pre-existing drugs. It can be assumed that the superexpression of *RPE65* in leiomyoma cells makes them sensitive to the synthetic retinol analogs used in pharmaceuticals for the treatment of acne: adapalene, tretinoin, isotretinoin, and tazarotene [75]. These products are indicated for topical use for the treatment of acne of mild and moderate severity, pilar keratosis, and other skin diseases.

## 6. Approaches to the Development of New Candidate Therapeutics for the Prevention and Treatment of Relapses of Leiomyoma, including Patients Preparing for Pregnancy or in the Process of Gestation

The development of new therapeutic agents should be started by testing those pharmaceutical substances that are already available for use and have passed preclinical clinical trials to the maximum extent (Table 3). Among those listed in Chapter 5, this category includes:(1)The immunobiological drugs secukinumab, ixekizumab, brodalumab, and netakimab, a recombinant humanized antibody against *IL17*—for all leiomyoma types;(2)Synthetic analogues of retinoids: adapalene, tretinoin, isotretinoin, and tazarotene—potential blockers of the RPE65 receptor—for all leiomyoma types;(3)Olaparib and niraparib—synthetic inhibitors of poly-ADP-ribosopolymerase (*PARP1*, also known as NAD+-ADP-ribosyltransferase 1 and *PARP2*)—for leiomyoma with biallelic inactivation of the fumarate hydratase gene as a driver mutation;(4)Resveratrol—for all leiomyoma types.

The development of inhibitors based on amino acid hydroxamates, carboxylates, thiols, phosphonates and sulfonamides, flavonoids, or 15oxycycline derivatives against matrix metalloproteinases ADAM12, *MMP11*, *MMP16*, and *THSD4* (Adamtsl6-β) should be considered a quick and easily accessible way to suppress their activity in vivo. The review by [75] contains a survey of the structure and functions of metalloproteinases. However, the work provides no data on the importance of these four proteinases in maintaining the normal functioning of human tissues and organs. In the work by [79], clinical trials of the following synthetic matrix protease inhibitors are reported: Marimastat (Pfizer, phase III, for the treatment of breast and lung cancer), Batimastat (British Biotech, phase II, for the treatment of malignant tumors), S-3304 (Shionogi, Phase II, for the treatment of solid lung cancer tumors), COL-3 (NSC-683551) (National Cancer Institute, phase I, refractory metastatic cancer), and CGS-27023A (Novartis, phase I, for the treatment of arthritis and malignancies).

## 7. Discussion

Since 1987, the development of medicines for the treatment of uterine leiomyoma and basic studies of the molecular mechanisms of the initiation of this disease were carried out in parallel without proper mutual coordination. For instance, goserelin acetate was patented in 1976 and approved by the FDA in 1983 [74]. Mifepristone clinical trials for the treatment of fibroids were completed in 2009 [40]. Cetrorelix acetate, described for the first time in 1997 [38], was allowed by the FDA for therapy of uterine leiomyoma in 2010. Ulipristal acetate, described for the first time in 2011 [41], has been under clinical trials since 2009 until now [45]. Clinical trials of asoprisnil were discontinued in 2005 at phase III due to registered endometrial changes in patients [82]. The last generation of protein kinase inhibitors of the pAKT/PIC3/mTOR signal pathway, capivasertib and ipatasertib, have not been clinically tested for therapy of uterine leiomyoma apparently due to obviously high side toxicity. This brief survey clearly demonstrates that recent achievements in meta-transcriptomics, meta-methylomics, and GWAS studies of uterine leiomyoma were almost not used for the development of novel therapeutics. Moreover, the firmly established metabolic differences between leiomyoma types caused by different driver mutations (misfunction of *MED12*, overproduction of *HMGA2*, and biallelic inactivation of *FH*) were not used for the development of specific medicines directed against specific targets of the different nosological forms of uterine leiomyomata except for relatively rare *FH*-dependent cases.

Analysis of the data by [62] suggests that leiomyomas with a driver mutation in the *MED12* gene should be considered a separate relevant object for the development of therapeutics. This follows from the greatest prevalence of this type of leiomyoma, as well as from the large number of target proteins in it, which are completely absent in normal tissues of the adult human body but necessary for the survival and growth of the tumor. This category should include, first of all, free and membrane-bound metalloproteinases: *ADAM12*, *MMP11*, *MMP16*, and *THSD4* (Adamtsl6-β). The development of specific inhibitors to them is a routine procedure, delivery to the target is facilitated by its extracellular location, and the absence of targets in normal tissues will ensure the absence of undesirable side effects. It also seems promising to create inhibitors of the potential-dependent channels *KCNAB3* (potassium-specific) and *CACNA1C* (calcium-specific), as well as hyaluronidases that stimulate cell migration (a product of the *KIAA1199* gene).

*MMP11* seems to be the most promising target for the development of target therapy of leiomyoma, since it is not expressed in any of the normal tissues of the body; however, a sustainable therapeutic effect of leiomyomata is expected only if the *MMP11* inhibitor is combined with blockers of some other targets. Trials of these inhibitors should be carried out mainly on patients with *MED12*-dependent leiomyomas. It can be assumed that these compounds will have low side toxicity and will be suitable for patients at all stages of pregnancy. Developing an inhibitor of pappalizin (*PAPPA2*), a protease that destroys IGFBP5 and IGF1 mRNA binding protein, a marker of osteoblasts, is also technically accessible. In this case, the trials should be carried out on a group of patients with *HMGA2*-dependent leiomyoma.

Creating inhibitors of membrane receptors, potential-dependent channels KCNAB3 and *CACNA1C*, is also a technologically easy task. Trials of these inhibitors should be carried out on patients with *MED12*-dependent leiomyoma.

A modern and effective way to remove unwanted proteins from the body (blocking their activity) is the development of immunobiological drugs based on humanized recombinant antibodies. An experimental approach allowing for the selection of targets for the development of such antibodies is presented in Figure 1. It is advisable to start their development with the antagonist proteins of the Wnt, Wif1, and Sfrp1 pathways, since they are secretory and universal in terms of the type of driver mutation that caused the development of the fibroid. The development of immunobiological drugs for the removal of *MMP11* and *MMP16* metalloproteases from the body also has good prospects. Their trials should be carried out on groups of patients with *MED12*-dependent leiomyomata. In addition, these drugs can be effectively used for the treatment of a number of malignant solid tumors, in particular, to prevent the metastasis of colon carcinoma. The prospects of this approach are confirmed by the data of [76], which provides the results of testing a DNA vaccine based on *MMP11* to protect mice from colon carcinoma. Immunization not only protected the animals from tumor progression but also did not show undesirable side effects.

In addition, an economically affordable way to protect patients against leiomyoma onset is the creation of preventive and protective vaccines based on proteins with high specificity for leiomyoma cells. An experimental approach allowing for the selection of targets for the development of vaccines coincides with one applicable for developing immunobiological preparations (humanized monoclonal antibodies) (Figure 1). A large set of markers can be attributed to this category: *Satb2, Nrg1, PCP4, CHRDL2*, and *MFAP2*—to combat all leiomyoma types; *HMGA2*, *GRPR, PLAG1, PAPPA2*, and *MB21D2*—to combat *HMGA2*-dependent leiomyoma; *ADAM12*, *MMP11*, *MMP16*, *RAD51B, THSD4* (Adamtsl6-β), and *RUNDC1*—to combat the *MED12*-dependent leiomyoma types. Vaccines can be either protein or genetic (based on DNA or RNA). Vaccines of the first type require longer development and testing but have a significantly lower manufacturing cost per dose, which makes them more accessible to patients. The great advantage of vaccination is safety for patients compared to chemical agents, which allows them to be used during pregnancy. The advantage of vaccines is also the possibility of the immunization of patients planning pregnancy with a high individual risk of leiomyoma onset, whereas the effect of immunization will be manifested during pregnancy, when the use of any medications is limited. An indirect argument confirming the effectiveness of vaccination as a means to combat the occurrence of leiomyomata is the fact that the risk of developing leiomyoma decreases significantly with each full-term pregnancy, whereas aborted pregnancies, on the contrary, increase the risk of the disease [3]. The reason for this, along with powerful hormonal effects occurring during pregnancy in the mother’s body, may be a factor of the immunization of the maternal organism with embryonic proteins, among which there are many antigens specifically expressed on the surface of leiomyoma cells, in particular: *Satb2, Nrg1, PCP4, CHRDL2, MFAP2*, *HMGA2*, *PLAG1, PAPPA2, MB21D2*, *ADAM12*, *MMP11*, and *MMP16*. The immunogenicity of the products of these proteins is probably not high by itself or due to the presence of the placental barrier, but due to repeated boosting during multiple pregnancies, the production of antibodies specifically suppressing the growth of the fibroids can gradually reach a significant level.

It is noteworthy that the effectiveness of the use of preventive vaccines to prevent the development of leiomyomas will depend on the development of diagnostic tools that allow for determining the individual risk of developing a certain type of leiomyoma in a patient. This task is currently still waiting for its resolution.

Finally, a promising area of leiomyoma therapy is the delivery of RNA and DNA constructs. For example, in the case of *HMGA2*-dependent leiomyomas, the delivery of miR-21, miR-23b, miR-29b, and miR-197 micro-RNAs described in [64], which suppress the expression of *HMGA2*. A total of 45 micro-RNAs with significantly increased or decreased content in the fibroids compared to the corresponding myometrium were identified.

A good effect can be expected from the delivery of miRNA to induce the specific degradation of the long non-coding RNA H19 described in [80]. This RNA is a universal marker of *MED12*- and *HMGA2*-dependent leiomyomata, participating in the regulation of the expression of the chromatin demethylation factor *TET3*.

*CEACAM1* (carcinoembryonic antigen-cell adhesion molecule 1), described by [42] and *KLF11* genes [74,78] should be suggested as candidate DNA moieties, the delivery of which to the leiomyoma cells in vivo is highly likely to contribute to their degradation due to the induction of endogenous apoptosis mechanisms as it happens in normal myometrium after a childbirth.

The production of RNA and DNA structures is a routine technological task, well-developed during the fight against the pandemic of the SARS-CoV-2 virus. However, for the targeted delivery of such molecules to the uterine leiomyoma cells, it is advisable to use functionalized vectors with specific affinity to the surface markers of these. In the case of *HMGA2*-dependent leiomyomata, *GPR20* and *IL11RA* can be specific receptors for solving this problem, since according to [62], their hyperproduction on cells of this type is observed. In the case of *MED12*-dependent leiomyomas, this function could be carried out by the membrane proteins *POPDC2, PLP1* (proteolipid protein, lipofilin, expressed in the central nervous system, responsible for the interaction of axon membranes with myelin), *THSD4* (Adamtsl6-β-transmembrane protein 4 with thrombospondine domain).

We anticipate that our study will promote the implementation of the data obtained by the modern high through-put molecular and bioinformatic methods for the development of efficient specific methods for mitigating the negative impact of leiomyomata on the life quality of the population.

## 8. Conclusions

The results of the metatranscriptomic and metamethylomic analyses suggest that the proteins *Satb2, Pcp4, Wif1*, *MMP11*, *MMP16*, and Adam12 are the most unique molecular markers expressed in all types of fibroids (with a driver mutation in the *MED12* gene, null-mutation in the *FH* gene, and with overexpression of the *HMGA2* gene), but silent in normal myometrium. These proteins should be investigated as candidate vaccine substances for the prevention of relapses of leiomyomatosis in patients with hereditary predisposition to this disease, particularly in those who are planning pregnancy. Additional candidate markers for the development of preventive and curative vaccines against leiomyomatosis can be identified as a result of immunological screening of blood sera from women who have given birth many times. Resveratrol and synthetic retinol analogs adapalene, tretinoin, isotretinoin, and tazarotene should be investigated as promising therapeutic agents that allow for slowing the growth of fibroids, including in patients with pregnancy.

## Figures and Tables

**Figure 1 diseases-11-00156-f001:**
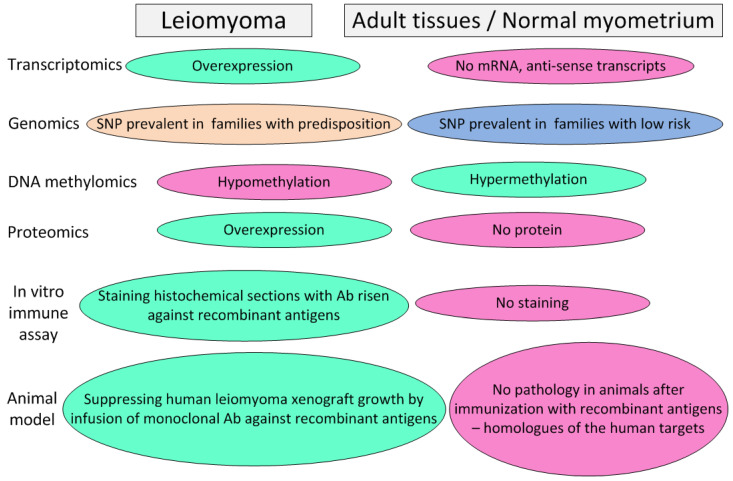
An experimental approach allowing for the selection of targets for the development of immunobiological preparations (humanized monoclonal antibodies) and vaccines for the prophylaxis and treatment of recurrent leiomyomata.

**Table 1 diseases-11-00156-t001:** Principal directions of uterine leiomyoma regulation studies and development of approaches to treatment of this condition.

Year of Publication, First Author	Reference	Key Results
Direction 1: *HMGA2* gene overexpression and chromosomal rearrangements in UL		
Nilbert et al., 1988	[19]	A total of 106 samples of leiomyoma biopsies were examined using classical cytogenetics methods. A normal parental karyotype was found in 57 samples (54%) capable of growing in culture, and chromosomal rearrangements were found in 20 samples (19%). In 10 cases (9%), they represented a translocation of the prethelomeric regions of chromosomes 12 and 14, and in 4 of them, there were other chromosomal rearrangements. In 10 cases where there was no translocation of the thelomeric region of chromosomes 12 and 14, minor rearrangements were observed in chromosomes 1, 2, 3, 4, 6, 8, 9, 10, 11, 13, and 19. Most often—in five cases, they affected chromosome 1.
Nilbert et al., 1992	[20]	It was established that with a variety of different combinations, only rearrangements of type t (1; 6) (q23;p21) and del (7) (q21.2q31.2) were systematically detected in independent samples and sufficient for the formation of a fibroid along with the most common rearrangements of type t (12; 14) (q14 - q15; q23 - q24).
Nilbert et al., 1990	[21]	Fibroids with complete trisomies on chromosome 12, with specific translocations t (12; 14) (q14 - 15; q23 - 24), were identified.
Hennig et al., 1997	[22]	It was discovered for the first time that the rearrangement involving the 12q14 - 15 prethelomeric region leads to an increase in the expression of the HMGIC gene, which later became known as *HMGA2*.
Klotzbücher et al., 1999	[23]	The expression of HMGIC (*HMGA2*) and HMGIY (HMGA1) genes in leiomyomas was studied using immunohistochemical staining of tissue sections. These authors reported that their expression of the genes of these non-histone proteins controlling the chromatin structure was observed in 16 of 33 samples of biopsies of the fibroids. At the same time, expression has never been observed in normal myometrium, as well as vascular endothelium and fibroblasts from the tumors. In 3 of the 16 biopsy samples showing HMGIC (*HMGA2*) expression, a protein product of this gene with an abnormal molecular weight was observed.
Klemke et al., 2009	[24]	The level of expression of the *HMGA2* gene was studied on a sufficient panel of samples from 180 patients. The highest levels of *HMGA2* expression were indeed observed in samples with rearrangements affecting the 12q14 - 15 region. But overexpression of *HMGA2* was repeatedly found in leiomyoma samples without such aberrations, although at lower levels.
Klemke et al., 2010	[25]	It was found that uterine leiomyomas are characterized by chromosomal rearrangements in the 12q14 - q15 region, leading to overexpression of the *HMGA2* gene. Recent studies have identified microRNAs of the let-7 family as post-transcriptional silencers of *HMGA2* expression. Chromosomal rearrangements sometimes lead to the appearance of shortened or hybrid *HMGA2* transcripts that lack 3’-UTR. The aim of the study was to use real-time RT-PCR to test how rearrangements of chromosomal region 12q14, leading to the appearance of shortened *HMGA2* transcripts in the fibroids, affect the stability of mRNA. The presented results prove that chromosomal rearrangements involving the *HMGA2* locus often lead to an increase in the mRNA lifetime, which contributes to overexpression.
Markowski 2010	[26]	Hormonal dependence of leiomyoma has long been known, and antitumor cytostatics are actively used in practice to suppress the growth of malignant neoplasms. The desire to use the PI3K/Akt/mTOR pathway as a therapeutic target is due to the fact that its shutdown does not cause immediate cell death, but leads to the so-called phenomenon of oncogen-induced aging, when cells under the action of proteins p16, p19, p53, and p21 gradually lose their division potential, weaken the antioxidant defense system, which weakens tumor aggressiveness with a relatively low level of side toxicity. Therefore, a balance between *HMGA2* and the p19Arf-TP53-CDKN1A axis was found to be essential for the growth of uterine leiomyomas.
Schoenmakers 2013	[27]	It is reported that repeated genomic rearrangements: del (7) (q22), t (12; 14) (q15; q24), t (1;2) (p36; p24), transpositions involving regions 6p21 and/or 10q22 occur in about 40% of the fibroids. These authors claim that in their previous works, they identified the genes HMGA1, *HMGA2*, RAD51L1, MORF, and NCOA1 as the primary targets of chromosomal rearrangements that cause the appearance of a benign tumor in each of the four variants of genome rearrangement using remote PCR methods.
Holzmann et al., 2014	[28]	The work reports that in the foci of leiomyomas that do not have somatic mutations in the *MED12* gene, chromotripsis phenomena were observed: numerous duplications and deletions of small segments were grouped mainly into five chromosomal regions: 2p14 - 2pter, 2q33.1 - 2q37.3, 5q31.3 - 5qter, 11q14.1 - 11qter and 18p11.21 - 18q2.3. Due to the small size of the rearranged fragments of genomic DNA, such rearrangements can hardly be detected by methods of classical cytogenetics. Histologically, the fibroids with chromotrypsin, as a rule, represent a cellular leiomyoma with pronounced hyperproduction of hyaluronic acid. The results of the work show that leiomyomas with a normal karyotype and without somatic mutations in the *MED12* gene are a heterogeneous group of diseases characterized by chromotripsis (“firestorm”), which does not affect the sites of chromosomal rearrangements characteristic of leiomyoma, such as 12q14 - q15 and 6p21.
Pradhan et al., 2016	[29]	The results of using the remote reverse PCR method for the detection and screening of de novo DNA rearrangements in uterine leiomyomas are reported. The method used makes it possible to identify genome rearrangements in the leiomyoma in comparison to the normal parental myometrium without putting forward an initial hypothesis about the location of recombination points. The screening of uterine leiomyoma samples for the presence of rearrangements in genomic locations allowed them to establish that the most susceptible to rearrangements of the genome in this type of tumor are located above the coding region of the *HMGA2* gene and inside the RAD51B gene. In particular, a previously undescribed point of genomic rearrangement above the *HMGA2* gene was identified, which went unnoticed in a previous study performed by genome-wide sequencing, where 30 samples of uterine leiomyoma showed no rearrangements within 1107 bp and 1996 bp analyzed in the RAD51B and *HMGA2* rearrangement hotspots.
Direction 2: Mutations in *MED12*		
Mäkinen et al. 2017	[30]	The authors of this work were the first to express the opinion that somatic mutations in the *MED12* gene, biallelic inactivation of the fumarate hydratase gene, and chromosomal aberrations leading to the overexpression of the *HMGA2* gene correspond to three mutually exclusive mechanisms of leiomyoma formation.
Wu et al., 2017	[31]	The work is devoted to elucidating the biological features of the two most common subtypes of uterine leiomyoma, mutant by *MED12* (*MED12*-LM) and overexpressing *HMGA2* (*HMGA2*-LM) uterine leiomyomas. Since each tumor carries only one genetic change, both subtypes are considered monoclonal. Approximately 90% of the cells in the *HMGA2* uterine leiomyoma were smooth muscle cells with overexpression of *HMGA2*. In contrast, *MED12*-LM consisted of the same number of smooth muscle cells and tumor-associated fibroblasts (TAFs). The TAFs did not carry mutations in *MED12*, which suggests an interaction between smooth muscle cells and fibroblasts, with different origins during the formation and growth of the fibroid.
Yin et al., 2015	[32]	Human uterine leiomyoma stem/progenitor cells expressing CD34 and CD49b initiate tumors in vivo.
Mas et al., 2015	[33]	Stro-1/CD44 as putative human myometrial and fibroid stem cell markers
Mu et al., 2016	[34]	IGF-1 and VEGF can be used as prognostic indicators for patients with uterine fibroids treated with uterine artery embolization.
Heikkinen et al., 2018	[35]	It was reported that there is a positive correlation between the increased expression of the COL3A1 gene in the fibroids and the expression of the HOXA13 gene, which is a regulator of the development of the organs of the female reproductive system, in particular, the cervix and vagina. According to these studies, a statistically significant increase in the expression of the HOXA13 gene above the level characteristic of normal myometrium was observed in both *MED12*-dependent and *HMGA2*-dependent fibroids.
Reis et al., 2016	[36]	Overexpression of COL4A1 and COL4A2 collagens in *MED12*-positive fibroids was detected.
Direction 3: Null-mutations in *FH* gene		
Sulkowski 2018	[37]	Krebs-cycle-deficient hereditary cancer syndromes are defined by defects in homologous recombination DNA repair
Mäkinen 2017	[30]	The authors of this work were the first to express the opinion that somatic mutations in the *MED12* gene, biallelic inactivation of the fumarate hydratase gene, and chromosomal aberrations leading to the overexpression of the *HMGA2* gene correspond to three mutually exclusive mechanisms of leiomyoma formation.
Direction 4: Agonists and antagonists of steroid hormones		
Gonzalez-Barcena et al., 1997	[38]	Treatment of uterine leiomyomas with luteinizing hormone-releasing hormone antagonist cetrorelix.
Nowicki et al., 2002	[39]	The influence of luteinizing hormone-releasing hormone analog on serum leptin and body composition in women with solitary uterine myoma.
Engman et al., 2009	[40]	Mifepristone for the treatment of uterine leiomyoma. A prospective randomized placebo-controlled trial.
Bouchard et al., 2011	[41]	Selective progesterone receptor modulators in reproductive medicine: pharmacology, clinical efficacy, and safety.
Malik et al., 2016	[42]	Gonadotropin-releasing hormone analogs inhibit leiomyoma extracellular matrix despite presence of gonadal hormones.
Patel et al., 2016	[43]	Mifepristone inhibits extracellular matrix formation in uterine leiomyoma.
Islam et al., 2021	[44]	Extracellular matrix and Hippo signaling as therapeutic targets of antifibrotic compounds for uterine fibroids.
Dababou et al., 2021	[45]	Linzagolix: a new GnRH antagonist under investigation for the treatment of endometriosis and uterine myomas.
Middelkoop et al., 2022	[46]	Evaluation of marketing authorization and clinical implementation of ulipristal acetate for uterine fibroids.
Arjona et al., 2022	[47]	The development of relugolix combination therapy as a medical treatment option for women with uterine fibroids or endometriosis is described.
Salas et al., 2022	[48]	New local ganirelix sustained release therapy for uterine leiomyoma. Evaluation in a preclinical organ model.
Chwalisz 2023	[49]	Clinical development of the oral gonadotropin-releasing hormone antagonist elagolix.
Direction 5: PI3K/Akt/mTOR and other intracellular signal pathway in UL		
Hu et al., 2009	[50]	Blockade of Wnt signaling inhibits angiogenesis and tumor growth in hepatocellular carcinoma.
Xu et al., 2014	[51]	It is hypothesized that the inhibition of AKT leads to the short-term triggering of specific mechanisms, which ultimately lead cells to cellular aging or death by the mechanism of apoptosis. It was experimentally shown that the inhibition of AKT leads to the accelerated aging of culture cells. The treatment of MK-2206 cells with an allosteric AKT inhibitor increased the content of reactive oxygen species, the level of miR-182 microRNA production, and the transcripts of several genes that are considered markers of ROS: p16, p53, p21, and β-galactosidase. The induction of ROS was associated with the hyperproduction of *HMGA2*, which was colocalized in the aging-related regions of heterochromatin.
Ye et al., 2014	[52]	Small-molecule inhibitors targeting activator protein 1 (AP-1).
Galindo LJ et al., 2018	[53]	Comparative analysis of AKT and the related biomarkers in uterine leiomyomas with *MED12*, *HMGA2*, and *FH* mutations.
Xie et al., 2018	[54]	The work is devoted to the study of the AKT signaling pathway and the mechanism of OIS in leiomyoma cells containing various driver mutations: *MED12* mutations (*n* = 25), *HMGA2* overexpression (*n* = 15), and biallelic inactivation of *FH* (*n* = 27). In each sample, the expression of genes involved in the response to sex steroids, the cell cycle, and the AKT pathway was studied by immunohistochemical method. It was found that the ER and PR genes were well expressed in all types of leiomyoma except for the *FH*-dependent type, which showed low ER expression and increased PR expression. *HMGA2*-dependent-type samples had significantly higher levels of AKT signaling and mitogenic activity than other types of the fibroids. *HMGA2* activated AKT signaling by enhancing IGF2BP2 expression. The suppression of HER2 expression in leiomyoma cells led to a decrease in AKT activity and an increase in the expression of p16 and p21, which ultimately caused oncogen-induced cell aging.
Alzahrani et al., 2019	[55]	The application of PI3K/Akt/mTOR inhibitors in cancer is described.
Pilgrim et al., 2020	[56]	Characterization of the role of activator protein 1 signaling pathway on extracellular matrix deposition in uterine leiomyoma.

**Table 2 diseases-11-00156-t002:** Principal directions of uterine leiomyoma regulation studies by GWAS, metatranscipromic, and metamethylomic approaches and using their data for development of candidate medicines for UL treatment based on small molecules, vaccines, and RNA.

Year of Publication, First Author	Reference	Key Results
Direction 1: GWAS for mapping polymorphisms associated with predisposition to UL onset		
Cha et al., 2011	[68]	GWAS identifies three loci associated with susceptibility to uterine fibroids.
Eggert et al. 2012	[69]	Genome-wide linkage and association analyses implicate FASN in predisposition to uterine leiomyomata.
Hellwege et al. 2017	[70]	A multi-stage genome-wide association study of uterine fibroids in African Americans.
Välimäki et al., 2018	[71]	These authors used genome-wide association analysis (GWAS) to identify genetic variants that are more common in people with fibroids. Using data from the British Biobank, the genomes of more than 15,000 women with fibroids were analyzed, which were compared with a control group of more than 392,000 individuals. The analysis revealed 22 regions of the genome, the genotypes of which differed in the experimental and control groups. These regions included genes that may well contribute to the development of fibroids, such as the TP53 gene, which affects the stability of the genome, and ESR1, which encodes the estrogen receptor (it is well-known that this hormone plays an important role in stimulating the growth of fibroids). Differences in genotypes were revealed for known genes involved in the control of the development of female genital organs.
Rafnar et al. 2018	[72]	Variants associated with uterine leiomyoma highlight genetic background shared among various cancers and hormone-related traits.
Edwards et al., 2019	[14]	*Trans-Ethnic Genome-Wide Association Study of Uterine Fibroids*
Gallagher et al., 2019	[73]	Genome-wide association and epidemiological analyses reveal common genetic origins between uterine leiomyomata and endometriosis.
Direction 2: Genome-wide transcriptome and DNA-methylome studies		
Wang et al., 2007	[64]	Paired samples of leiomyomas and normal myometrium from 41 patients were examined, which were used to construct banks of micro-RNA and their subsequent sequencing. As a result of bioinformatic analysis, 45 microRNAs with significantly increased or decreased content in the fibroids compared to the corresponding myometrium were identified (*p* < 0.001). The five undergoing the strongest expression change are the let-7 family: miR-21, miR-23b, miR29b, and miR-197.
Navarro et al., 2012	[61]	The objective of the work was to identify abnormally methylated sections of the genome in UL cells in vivo using genome-wide analysis methods and to compare the data obtained with the results of the metatranscriptome analysis. Biological materials in the form of paired samples of leiomyomas and adjacent normal myometrium were selected from 18 patients of African-American origin. A total of 55 genes with differential methylation of the promoter regions were identified, which correlated with differences in the level of expression in uterine leiomyoma compared to normal myometrium. Additionally, 80% of the identified genes showed an inverse relationship between the status of DNA methylation and the content of the corresponding mRNA in uterine leiomyoma tissues, including 34 genes which demonstrated hypermethylation of the promoter region and a corresponding decrease in expression level, and 10 genes which demonstrated demethylation and an increase in expression level.
Mehine et al., 2016	[62]	The data of a meta-transcriptomic study of the expression profile of leiomyomas with four types of driver mutations in comparison to the expression profile of the adjacent normal myometrium, are presented. A total of 19 upregulated and a single downregulated markers of leiomyomata (in comparison to the normal myometrium) are reported.
Anjum et al., 2019	[63]	The levels of expression of marker transcripts in UL were measured using meta-transcriptomic analysis on the Illumina platform, followed by validation of the most significant results using real-time RT-PCR. A full transcriptome analysis showed an increase in the expression of 128 genes in the fibroids compared to normal myometrium and a decrease in the expressions of 98 genes.
George et al., 2019	[65]	Integrated epigenome, exome, and transcriptome analyses were carried out for revealing the molecular subtypes of fibroids. Cases of HER2 overexpression-independent chromosomal rearrangement were found. Hypomethylation of the structural region of the *HMGA2* gene was found to be a reason for the overexpression. It was found that *MED12* mutations and increased *HMGA2* expression can coexist in the same fibroids. Increased expression of insulin receptor substrate undergoing phosphorylation when binding insulin to the receptor (IRS4) in leiomyoma cells compared to myometrium was found.
Maekawa et al., 2022	[57]	Differences in the DNA methylome, transcriptome, and histological features in uterine fibroids with and without *MED12* mutations were studied. Genes of inflammatory response (CCL2, AOX1, ACKR1), apoptosis (ANXA1, CITED2), and metabolism associated with reactive oxygen species were reported to undergo hypermethylation in all types of leiomyoma cells. It was found that 80–90% of the fibroids of all types showed overexpression of the SATB2 and NRG1 genes compared to the myometrium, and the excess level varied from 1.5 to 20–30 times.
Direction 3: Candidate medicines for UL treatment based on small molecules, vaccines, and RNA		
Walker et al., 1983	[74]	*Therapeutic potential of the LHRH agonist, ICI 118630, in the treatment of advanced prostatic carcinoma*
Hinterhuber et al., 2005	[75]	*Expression of RPE65, a putative receptor for plasma retinol-binding protein, in nonmelanocytic skin tumours*
Peruzzi et al., 2009	[76]	*MMP11 is described as a novel target antigen for cancer immunotherapy.*
Yin et al., 2010	[77]	*Transcription factor KLF11 integrates progesterone receptor signaling and proliferation in uterine leiomyoma cells*
Zheng et al., 2014	[78]	*Epigenetic regulation of uterine biology by transcription factor KLF11 via posttranslational histone deacetylation of cytochrome p450 metabolic enzymes*
Ho et al., 2018	[67]	*Resveratrol inhibits human leiomyoma cell proliferation via crosstalk between integrin avß3 and IGF-1R*
Grigorkevich et al., 2019	[79]	*Matrix metalloproteinases and their inhibitors*
Cao et al., 2019	[80]	*H19 lncRNA has been identified as a master regulator of genes that control uterine leiomyomas*

**Table 3 diseases-11-00156-t003:** Candidate targets for the development of uterine leiomyoma therapeutic agents.

Nosological Type of the Tumor	Chemical Substances	Humanized Monoclonal Antibodies	Vaccines	RNA Delivery
*MED12*-dependent	*ADAM12, MMP11, MMP16, KCNAB3, CACNA1C, RAD51B*	*ADAM12, MMP11, MMP16, RUNDC1*	*ADAM12, MMP11, MMP16, RUNDC1, RAD51B*	miR-200c, miR-93
*HMGA2*-dependent	*PAPPA2, MB21D2*,	*GRPR, PLAG1, PAPPA2, MB21D2*	*HMGA2, PLAG1, PAPPA2, MB21D2*	miR-21, miR-23b, miR-29b, miR-197, mir-106b
*FH*-dependent	*PARP* (olaparib *, niraparib *), *TNFRSF21, NQO1, SLC7A11, FAM46C, ABCC3*	-	-	-
All types together	Resveratrol **, *RPE65*: adapalene **, tretinoin **, isotretinoin **, and tazarotene **	*IL17 *, WIF1, SFRP1, SATB2, NRG1, PCP4, CHRDL2, MFAP2*	*PCP4, CHRDL2, MFAP2*	Anti-H19, miR-182

* Substance is approved for treatment of some conditions besides leiomyoma; ** substance is used as a para-pharmaceutical.

## Data Availability

No Supplementary Materials are available.

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
