# Peer review of "Novel Approaches to Possible Targeted Therapies and Prophylaxis of Uterine Fibroids"

_diseases, 2023, doi:10.3390/diseases11040156_

Round 1

Reviewer 1 Report (Previous Reviewer 3)

Comments and Suggestions for Authors

The manuscript is an extensive review of everything in the literature on fibroids and molecular studies on them. It is a complete and exhaustive overview of the molecular patterns of fibroids and the possible pharmacological treatments to be used based on chemical compounds, humanized recombinant antibodies, vaccines based on markers of the uterine leiomyoma cells that are absent in the adult organism, DNA and RNA preparations. Obviously these are futuristic speculations, not yet feasible, but the review is interesting and can provide inspiration for further studies.

Author Response

Dear Reviewer, thank you so much for your efforts and for the positive evaluation of our MS. Best regards, sincerely yours Elena

Reviewer 2 Report (Previous Reviewer 2)

Comments and Suggestions for Authors

As I already stated having opportunity to get familiar with revised paper:

"The manuscript is well written and covers all important issues concerning uterine fibroids. In regard to very wide analysis of available literature it seems to sufficiently explains evaluated data. Personally I’m not enthusiastic with the usage “uterine nodes” term however we may leave it as it is clear… there are some typo, editorial flaws in the references to be fixed (missing some numbers). I would also go to the table 1  and check if the chronology of cited papers is satisfactory. As last point in such deep analyses important and very valuable for the readers are schemes or graphs which are not used in revised manuscript. I would suggest to introduce them. Potentially pathways are the best idea with pointed targets. "

I cannot compare exactly the paper previous version and present, however I assumpt that my cosmetic suggestions were not taken into account, sadly. 

Author Response

Dear Reviewer,

Thank you so much for your high evaluation of our MS and for the advices. We checked MS for typos and introduced the following revisions:

  1. The term “uterine nodes” is changed for “fibroids” along the whole text.
  2. Fig. 1 ‘An experimental approach allowing to select targets for the development of the immunobiological preparations (humanized monoclonal antibodies) and the vaccines for prophylaxis and treatment of the recurrent leiomyomata’ is added in order to highlight our idea about development of the therapeutic antibodies and vaccines against leiomyomas.

Tank you again so much!

Best regards,

Sincerely yours Elena

This manuscript is a resubmission of an earlier submission. The following is a list of the peer review reports and author responses from that submission.

Round 1

Reviewer 1 Report

Comments and Suggestions for Authors

I congratulate the authors for their work. There would be a few minor issues to address:

1. In the title please mention that this manuscript is a review.

2. Include a methods section in which you mention:

-the eligibility criteria of the studies

- databases consulted

- search strategy.

Reviewer 2 Report

Comments and Suggestions for Authors

The manuscript is well written and covers all important issues concerning uterine fibroids. In regard to very wide analysis of available literature it seems to sufficiently explains evaluated data. 
Personally I’m not enthusiastic with the usage “uterine nodes” term however we may leave it as it is clear… there are some typo, editorial flaws in the references to be fixed (missing some numbers). I would also go to the table 1  and check if the chronology of cited papers is satisfactory. As last point in such deep analyses important and very valuable for the readers are schemes or graphs which are not used in revised manuscript. I would suggest to introduce them. Potentially pathways are the best idea with pointed targets. 

Reviewer 3 Report

Comments and Suggestions for Authors

I've already reviewed the same work for two journals, with the same result.

This work is fine as a book chapter, but not as a journal article.

It is a long review already written and rewritten in other journals, nothing new, but the umpteenth representation of what has been known and known for years.

Therefore the result of my review is to reject the manuscript and submit it as a book chapter.

Comments on the Quality of English Language

Fine